

# Primary productivity as a control over soil microbial diversity along environmental gradients in a polar desert ecosystem

Kevin M. Geyer[1], Cristina D. Takacs-Vesbach[2], Michael N. Gooseff[3,4] and John E. Barrett[5]

[1] Department of Natural Resources and the Environment, University of New Hampshire, Durham, NH, USA
[2] Department of Biology, University of New Mexico, Albuquerque, NM, USA
[3] Institute of Arctic and Alpine Research, University of Colorado, Boulder, CO, USA
[4] Department of Civil, Environmental and Architectural Engineering, University of Colorado, Boulder, CO, USA
[5] Department of Biological Sciences, Virginia Tech, Blacksburg, VA, USA

Corresponding author
Kevin M. Geyer,
kevin.geyer@unh.edu

## ABSTRACT

Primary production is the fundamental source of energy to foodwebs and ecosystems, and is thus an important constraint on soil communities. This coupling is particularly evident in polar terrestrial ecosystems where biological diversity and activity is tightly constrained by edaphic gradients of productivity (e.g., soil moisture, organic carbon availability) and geochemical severity (e.g., pH, electrical conductivity). In the McMurdo Dry Valleys of Antarctica, environmental gradients determine numerous properties of soil communities and yet relatively few estimates of gross or net primary productivity (GPP, NPP) exist for this region. Here we describe a survey utilizing pulse amplitude modulation (PAM) fluorometry to estimate rates of GPP across a broad environmental gradient along with belowground microbial diversity and decomposition. PAM estimates of GPP ranged from an average of 0.27 $\mu$mol $O_2$/m$^2$/s in the most arid soils to an average of 6.97 $\mu$mol $O_2$/m$^2$/s in the most productive soils, the latter equivalent to 217 g C/m$^2$/y in annual NPP assuming a 60 day growing season. A diversity index of four carbon-acquiring enzyme activities also increased with soil productivity, suggesting that the diversity of organic substrates in mesic environments may be an additional driver of microbial diversity. Overall, soil productivity was a stronger predictor of microbial diversity and enzymatic activity than any estimate of geochemical severity. These results highlight the fundamental role of environmental gradients to control community diversity and the dynamics of ecosystem-scale carbon pools in arid systems.

## INTRODUCTION

Primary production plays a fundamental role in controlling terrestrial foodwebs by making available the resources that regulate consumer productivity (*Lindeman, 1942*; *Tilman, 1982*; *McNaughton et al., 1989*) and shape community diversity (*Waide et al., 1999*;

*Judd, Crump & Kling, 2006*). Rates of primary production also reflect the geochemical suitability of habitats for soil organisms (*Chapin, 1980*), and thus much valuable information about the abiotic and biotic components of soil ecosystems can be inferred through knowledge of carbon fixation rates. Arid soils like those of the McMurdo Dry Valleys of Antarctica exemplify the tight coupling between ecosystem process rates and soil biological/geochemical properties. Here the landscape is dominated by alkaline (pH > 9.0), saline (conductivity >500 $\mu$S/cm), dry (gravimetric moisture < 1%), and low organic matter (<0.03% organic C by weight) soils (*Barrett et al., 2004*) that support a very limited diversity of microfauna and no vascular plant or vertebrates species (*Adams et al., 2006*). However, seasonal wetting of stream and lake margins in this polar desert ameliorates the environmental severity and fosters dense cryptogamic mats of cyanobacteria and moss (*Barrett, Gooseff & Takacs-Vesbach, 2009*). Such primary production reinforces the habitability of these hotspots for diverse microbial organotrophs through amendments of organic carbon (*Treonis, Wall & Virginia, 1999*; *Simmons et al., 2009*; *Geyer et al., 2013*).

Despite a long history of researching environmental controls over dry valley soil diversity (*Virginia & Wall, 1999*; *Barrett et al., 2006b*; *Cary et al., 2010*; *Lee et al., 2012*), efforts to measure rates of terrestrial gross or net primary production (GPP, NPP) in this system have been few. Rates of carbon fixation are near the levels of instrumental detection for gas exchange techniques and the growth rate of cryptogams does not lend itself to growth-increment based methods. Notable exceptions include the pioneering work of E. Imre Friedmann on cryptoendolithic lichens (0.6 g C/m$^2$/y NPP) (*Friedmann et al., 1993*) and more recent surveys of coastal moss turfs (250 g C/m$^2$/y NPP) (*Pannewitz et al., 2005*) and riparian *Nostoc* cyanobacterial mats (20 g C/m$^2$/y NPP) (*Novis et al., 2007*). The availability of new approaches to *directly* measure photosynthetic activity (e.g., pulse amplitude modulation (PAM) fluorometry) provides a unique opportunity to compare productivity estimates among analytical approaches and refine our understanding of the energetic basis of dry valley foodwebs.

Here we present the results of a field survey in the McMurdo Dry Valleys where the important ecological functions of productivity and decomposition were examined along a broad environmental gradient previously demonstrated as an important driver of biotic diversity and activity (*Geyer et al., 2013*). PAM fluorometry was used to estimate primary productivity while exoenzyme activity assays provide an indication of detrital pathways and the diversity of organic substrates. We discuss the overall significance of primary production within a low organic matter ecosystem toward influencing subsurface processes and community structure, as well as promising avenues for using PAM fluorometry in conjunction with other techniques to constrain rates of primary production (and its sensitivity to environmental factors) within arid systems.

## SITE DESCRIPTION AND METHODS

### Site description

Surveys were conducted in Taylor Valley, a region well characterized by previous research within the larger McMurdo Dry Valleys, Antarctica. Soils of this region are a poorly weathered, dry-permafrost composed of >90% sand-sized particles with ice cement

occurring within 0.5 m of the surface (*Ugolini & Bockheim, 2008*). Salinity and pH are generally high, a consequence of limited vertical water movement through soil layers that results in the accumulation of weathered carbonates and aerially deposited salts (*Bockheim, 1997*). Low temperatures restrict photosynthesis within this region to an approximately 6–8 week austral summer period when 24 h radiation elevates air and soil surface temperature (10 and 25 °C maxima, respectively) (*Doran et al., 2002*) and stimulates the melting of ice and snow to yield free water. Cryotolerant organisms such as cyanobacterial mats are reactivated and, in some cases, can resume photosynthesis and nitrogen fixation within minutes of rehydration along the soil surface (*Vincent & Howard-Williams, 1986*; *McKnight et al., 2007*). A surprisingly diverse assemblage of microbes exists underground alongside a limited variety of metazoan invertebrates at higher trophic levels (*Adams et al., 2006*; *Takacs-Vesbach et al., 2010*).

Communities of cyanobacteria, moss, lichens, and eukaryotic algae are responsible for primary production in this region and are often characterized by their niche habitat. For instance, a diversity of lithophytic cyanobacteria (e.g., families *Nostocaceae* and *Oscillatoriaceae*) are commonly divided into the operational categories hypolithic, endolithic, and epilithic according to the rock surface colonized (*Broady, 1996*; *Pointing et al., 2009*). Cyanobacteria and fewer than ten species of moss frequently form dense cryptogamic mats along the wetted soil margins of streams, lakes, and snowpacks (*Seppelt & Green, 1998*). A limited number of distinguishing morphological characteristics makes in-field identification challenging, and thus most mat-forming colonies are identified by morphotypes of color (e.g., black, orange, red) and/or physical location (e.g., wetted stream margin, submerged aquatic) (*Broady, 1996*; *McKnight et al., 2007*). The dominant soil bacterial phyla in this region include many of the more common groups found worldwide, such as *Actinobacteria*, *Acidobacteria*, and *Bacteroidetes* (*Cary et al., 2010*).

Previous field surveys during the austral summer of 2010/2011 from Taylor and neighboring Wright Valleys indicated that soils associated with an a priori productivity gradient (defined by surface density of microbial mats) ranged over three orders of magnitude in chlorophyll *a* concentrations (0.30–270 µg/g dry material). Across this gradient was observed an increase in soil moisture, organic carbon, invertebrate abundance, microbial biomass carbon (MBC), and diversity of bacteria (*Geyer et al., 2013*; *Ball & Virginia, 2014*). The activity of two common carbon-acquiring microbial exoenzymes (α- and β-glucosidase) was also positively associated with soil productivity, suggesting that productive habitats are more decompositionally active because of greater organic substrate concentrations, enhanced activity or biomass of decomposers, or perhaps both.

## Soil sampling

Here we report the results of continued sampling along this productivity gradient performed during the austral summer (January 2013) using five broad regions (Fig. 1). These regions ranged from tens of meters to tens of kilometers apart and consisted of common soil habitats found in Taylor Valley such as stream margins, the wetted edge of

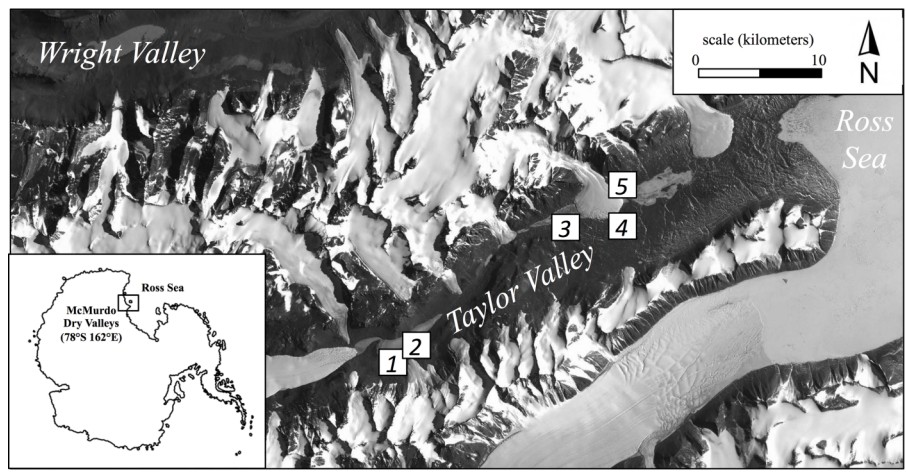

**Figure 1** Location of the five regional sampling sites in Taylor Valley of the McMurdo Dry Valleys, Antarctica.

**Table 1 Description of the five sampling regions from which three locations (each) were chosen to collect samples.** Locations within each region were chosen to capture the range of soil primary productivity visually apparent.

| Region | Landscape location | Latitude/longitude (decimal degree) | Elevation (meters above sea level) |
|---|---|---|---|
| 1 | Bonney Riegel, near Wormherder Creek | −77.733333/162.320183 | 294.5 |
| 2 | Bonney Riegel, near Wormherder Creek | −77.730383/162.334400 | 259.9 |
| 3 | Snowpack margin, near south shore Lake Hoare | −77.637333/162.881200 | 151.0 |
| 4 | Upper Green Creek margin | −77.624400/163.05403 | 18.1 |
| 5 | Canada Stream margin | −77.615417/163.041450 | 42.9 |

snowpacks, and hyperarid soils (Table 1; Fig. 1). Although specific cryptogam identification was not determined, moss and/or cyanobacteria mats frequently colonized more productive soils but appeared absent in others. Three locations (2.5 m$^2$ each) were chosen to capture the range of visually apparent surface production within each of the five regions (15 locations overall); in so doing, soils along a productivity gradient were collected from both within each region and across the greater Taylor Valley. Locations of high productivity contained dense cryptogam mats up to 5 cm thick, while arid soils appeared barren and without conspicuous surface producers (Fig. S1). PAM measurements were made on light-adapted surfaces at eight equidistant points within each location following a gridded pattern. Triplicate soils were collected from within each location such that surface cryptogams were stored separately as replicates and subsurface soils (to a depth of 5 cm per pit) combined to produce one composite sample (~500 g) per location. Ten grams of soil from this composite sample was immediately preserved in a sucrose-lysis buffer for nucleic acid stabilization (*Mitchell & Takacs-Vesbach, 2008*). All samples were frozen at −20 °C (molecular samples −80 °C) within 48 h of collection. Field sampling was permitted under McMurdo LTER NSF OPP grant 1115245.

## Biogeochemical and molecular analyses

Surface cryptogam biomass was measured indirectly as chlorophyll *a* concentrations via spectrophotometry from the acetone extract of dried surface soils (*Castle, Morrison & Barger, 2011*; *Geyer et al., 2013*). Subsurface soil was 2 mm sieved and used for all subsequent analyses. Soil pH and electrical conductivity were measured from a 1:2 and 1:5 soil/water slurry, respectively, using standard procedures developed for this region (*Nkem et al., 2006*). Soil water content was determined gravimetrically by oven-drying for 48 h at 105 °C. Total nitrogen (TN) was estimated from ~300 mg of ground, dried, and acidified soil using a FlashEA 1112 NC Elemental Analyzer (CE Elantech, Lakewood, NJ, USA). Chloroform-labile carbon was used as an indication of soil MBC where soil samples were fumigated with gaseous chloroform for five days under vacuum (*Cheng & Virginia, 1993*). Paired fumigated and non-fumigated samples were then extracted with a 0.5 M $K_2SO_4$ solution and final extracts analyzed for total organic carbon using a OI Model 1010 Total Organic Carbon Analyzer (OI Analytical, College Station, TX, USA), where final chloroform-labile carbon was calculated as the difference between fumigated and non-fumigated total organic carbon. Non-fumigated extracts were used as estimates of soluble soil organic carbon (SOC).

Potential soil extracellular enzyme activity was assayed for five carbon and nitrogen acquiring enzymes (Table 2) to characterize the diversity and magnitude of hydrolytic and oxidative decompositional pathways (*Sinsabaugh & Shah, 2011*). These measures were also examined as an index of organic matter complexity (*Tscherko et al., 2003*). Hydrolytic activity was measured using 0.5 g soil incubations in the presence of labeled substrates and 50 mM $NaHCO_3$ buffer (pH = 8.2) following the methods of *Zeglin et al. (2009)*. Oxidative assays underwent similar treatment, although standards were created by reacting a known mass of L-3,4-dihydroxyphenylalanine (L-DOPA) with horseradish peroxidase from which a standard curve (dilution series) of the product was used to infer activity within field samples. For all assays, triplicate samples were incubated at room temperature on a platform shaker (250 rpm) for a minimum of 2 h and enzyme-induced fluorescence (hydrolytic enzymes) measured by excitation (365 nm) and emission (450 nm) or light absorbance (oxidative enzyme) measured by absorbance (460 nm) using a Tecan Infinite M200Pro plate reader (Tecan, Mannedorf, Zurich, Switzerland). In addition to sample incubations, control (buffer only), substrate (substrate + buffer), and standard (standard + buffer) references were analyzed to account for other sources of fluorescence. Final activity was expressed as activity (nanomole of substrate cleaved) per hour gram per MBC. Organic substrate breakdown is assumed to be entirely the result of microbially exuded enzymes.

Bacterial diversity was estimated using a terminal restriction fragment length polymorphism (TRFLP) procedure, a conservative estimate of phylum-level diversity (*Thies, 2007*). DNA was extracted from soils using a modified cetyltrimethylammonium bromide (CTAB) procedure involving a mixture of 1% CTAB, 10% sodium dodecyl sulfate, phenol/chloroform/isoamyl alcohol (pH = 7.5), lysozyme (0.2 $\mu$g/$\mu$L), and proteinase K (20 $\mu$g/$\mu$L) with ~0.75 g soil. PCR amplification took place in triplicate

**Table 2 Additional information for enzymatic assays of soils.** Standards for the phenol oxidase assay were created by reacting a known mass of L-3,4-dihydroxyphenylalanine substrate with a horseradish peroxidase.

| Enzyme | Shorthand | Activity | Substrate | Standard | Target |
|---|---|---|---|---|---|
| α-Glucosidase | AG | Hydrolytic | 4-MUB-α-D-glucopyranoside | 4-MUB | Starch |
| β-Glucosidase | BG | Hydrolytic | 4-MUB-β-D-glucopyranoside | 4-MUB | Cellulose |
| N-Acetyl-β-glucosaminidase | NAG | Hydrolytic | 4-MUB-N-acetyl-β-D-glucosaminide | 4-MUB | Chitin |
| Phenol oxidase | POX | Oxidative | L-3,4-Dihydroxyphenylalanine | N/A | Lignin |
| Leucine aminopeptidase | LAP | Hydrolytic | L-Leucine-7-amido-4-methylcoumarin HCl | 7-Amino-4-methylcoumarin | Protein |

**Note:**
4-MUB, 4-methylumbelliferyl.

using a standard 2 $\mu$L of diluted template, 0.025 units/$\mu$L of Taq Hot Start Polymerase (Promega Corporation, Madison, WI, USA), and the universal bacterial primers 8F (5′-AGAGTTTGATCMTGGCTCAG-3′) and 519R (5′-ACCGCGGCTGCTGGCAC-3′), the forward primer labeled with a 5′ 6-FAM fluorophore (Integrated DNA Technologies, Coralville, IA, USA). Amplification reaction conditions were previously optimized for these soils by *Geyer et al. (2013)*. Successful amplifications (13 of 15 samples) were digested with *Hae*III (New England BioLabs, Ipswich, MA, USA) in triplicate for 3 h at 37 °C following manufacturer's suggested protocols. Fragment separation/ quantification took place in quadruplicate with an ABI 3130xl Genetic Analyzer (Applied Biosystems, Carlsbad, CA, USA) and fragments binned using the GeneMarker software AFLP protocol. Resulting sample profiles were standardized using the procedures outlined by *Dunbar, Ticknor & Kuske (2001)* to produce both a consensus profile among replicates and final normalization of all sample profiles by total sample fluorescence.

## PAM fluorometry

A MINI-PAM (Walz) PAM fluorometer was used to examine rates of surface cryptogam production in situ for 12 locations. The PAM fluorometer uses saturating light to induce a measurable change in fluorescence directly proportional to the drop in photochemical quenching which results from the instantaneous light-induced reduction of the photosystem II (PSII) electron transport chain. The key measurements obtained from the PAM were effective quantum yield of PSII (YII) and electron transport rate (ETR). Additional measures of photon flux density (photosynthetically active radiation, or PAR) and temperature at the cryptogam surface were also recorded. YII is the proportion of incident light used to drive the photochemistry of photosynthesis (*Ritchie & Bunthawin, 2010*), while ETR is derived from the product of YII, PAR, and two factors which account for a photon allocation factor between PSI and II (0.5) and a mean absorptance factor (0.84) previously described for a variety of plants.

$$ETR(\mu mol/m^2/s) = YII \times PAR \times 0.5 \times 0.84$$

Electron transport rate is thus an estimate of the rate of electron passage through PSII. Because four electrons pass through PSII per oxygen molecule produced during photosynthesis, estimates of gross photosynthesis (or more specifically, photosynthetic

capacity) were calculated using (*Figueroa, Conde-Alvarez & Gomez, 2003*; *Ritchie & Bunthawin, 2010*):

$$\text{GPP}(\mu\text{mol O}_2/\text{m}^2/\text{s}) = 1/4 \times \text{ETR}$$

A strong linear relationship between ETR and gross photosynthesis has been previously demonstrated for Antarctic mosses (*Green et al., 1998*, *Masojidek et al., 2001*) and has been extrapolated to the cryptogams surveyed here. Being poikilohydric, the photosynthetic activity of Antarctic cryptogams is further constrained by moisture availability (*Schroeter et al., 2011*). Our seasonal estimates of GPP should thus be interpreted as simplifications that will require higher resolution moisture, temperature, and light conditions in order to be refined. All PAM measurements were auto-corrected for background fluorescence of non-biological material (e.g., rocks).

## Data analysis

Data analysis (correlation, *t*-tests, ANOVA, and simple/multiple linear regressions) was performed using SAS JMP. The normality of studentized residuals was examined and, if found significantly non-normal by Shapiro–Wilk's test, either $\log_{10}$ (e.g., all enzyme activity values) or square-root transformed. Locations were clustered into three conservative productivity classes (e.g., low, intermediate, high) based on a *k*-means non-hierarchical clustering technique of soil moisture. Spearman correlation was used to assess pairwise interactions between all variables because of the presence of some nonlinear (monotonic) relationships. Multiple regression was performed using mixed stepwise selection of model parameters ($\alpha = 0.15$) that had variance inflation factors <10. Both TRFLP fragment abundances and enzyme activities were used in calculations of Shannon–Wiener diversity indexes. Enzyme diversity (Enz. H′) calculations include the activity of α-glucosidase (AG), β-glucosidase (BG), *N*-acetyl-β-glucosaminidase (NAG), and phenol oxidase (POX) (Table 2) to provide a metric of carbon-acquiring enzyme activity from which the diversity of carbon substrates was inferred.

## RESULTS

Soil characteristics of the 15 locations sampled resemble those observed in previous work conducted in hydrological margins of streams and lacustrine environments of the McMurdo Dry Valleys (*Barrett, Gooseff & Takacs-Vesbach, 2009*; *Geyer et al., 2013*) and captured gradients of gravimetric water content ranging over an order of magnitude (range = 1.3–17.3%; mean = 11.4%) and chlorophyll *a* concentrations over two orders of magnitude (range = 0.02–2.84 μg/g dry material; mean = 0.88 μg/g dry material). Molar C:N of total soil (range = 6.6–11.3; mean = 8.3) did not vary significantly with other measures of soil productivity. Correlational trends indicate that soil properties associated with physicochemical severity (e.g., electrical conductivity and pH) are positively correlated with one another, yet negatively correlated with properties of mesic, productive soils like chlorophyll *a*, microbial biomass carbon, and bacterial (TRFLP) diversity (Table 3). Because of its importance as a driver of most soil parameters in this arid system, soil moisture was chosen as a basis for clustering locations into three

**Table 3 Spearman correlation matrix for soil properties.**

| Variable | EC | Chl*a* | % Moist | SOC | TN | MBC | Bact. H′ | AG | POX | Enz. H′ | ETR |
|---|---|---|---|---|---|---|---|---|---|---|---|
| pH | 0.61* | −0.36 | −0.69** | −0.36 | −0.38 | −0.55* | 0.039 | −0.38 | 0.56* | −0.52* | −0.16 |
| EC | | −0.25 | −0.62* | −0.39 | −0.38 | −0.34 | −0.26 | −0.54* | 0.33 | −0.59* | −0.63* |
| Chl*a* | | | 0.80*** | 0.50 | 0.55* | 0.47 | 0.57* | 0.52* | −0.57* | 0.66** | 0.48 |
| % Moist | | | | 0.40 | 0.44 | 0.49 | 0.57* | 0.62* | −0.52* | 0.68** | 0.50 |
| SOC | | | | | 0.99*** | 0.79*** | 0.13 | 0.59* | −0.84*** | 0.61* | 0.40 |
| TN | | | | | | 0.80*** | 0.19 | 0.61* | −0.86*** | 0.86*** | 0.40 |
| MBC | | | | | | | 0.00 | 0.30 | −0.89*** | 0.62* | 0.11 |
| Bact. H′ | | | | | | | | 0.57* | −0.03 | 0.29 | 0.57 |
| AG | | | | | | | | | −0.44 | 0.83*** | 0.78** |
| POX | | | | | | | | | | −0.77*** | −0.05 |
| Enz. H′ | | | | | | | | | | | 0.63* |

Notes:
EC, Electrical conductivity (μS/cm); Chl*a*, chlorophyll *a* (μg/g dry soil); Moist, gravimetric moisture (%); SOC, soil organic carbon (mg/kg dry soil); TN, total nitrogen (mg/kg dry soil); MBC, microbial biomass carbon (mg/kg dry soil); Bact. H′, TRFLP bacterial diversity; AG, α-glucosidase activity (nmol/g MBC/h); POX, phenol oxidase activity (nmol/g MBC/h); Enz. H′, diversity index of activity for carbon-acquiring enzymes; ETR, electron transport rate (μmol/m²/s).
* $p < 0.05$.
** $p < 0.01$.
*** $p < 0.001$.

**Table 4 Average (untransformed) edaphic properties for 15 soil habitats clustered by three productivity zones.**

| Variable | Productivity zone | | |
|---|---|---|---|
| | Low ($n = 3$) | Intermediate ($n = 7$) | High ($n = 5$) |
| pH | 8.77a (0.04) | 8.75a (0.04) | 8.5b (0.07) |
| EC | 99.57a (35.76) | 54.6ab (7.49) | 26.56b (6.14) |
| Chl*a* | 0.08a (0.05) | 0.73ab (0.23) | 1.56b (0.39) |
| % Moist | 3.94a (1.28) | 11.51b (0.73) | 15.85c (0.56) |
| SOC | 234.15a (29.87) | 418.03a (82.88) | 438.71a (104.03) |
| TN | 34.66a (3.98) | 55.34a (9.71) | 61.00a (13.26) |
| MBC | 7.44a (2.23) | 12.93a (2.66) | 15.57a (3.49) |
| Bact. H′ | 4.06a (NA) | 4.23a (0.05) | 4.27a (0.05) |
| AG | 6005a (1987) | 9783a (2279) | 14767a (3505) |
| BG | 7233a (988) | 15048ab (3020) | 25194b (3621) |
| NAG | 1633a (415) | 2028a (272) | 2502a (316) |
| POX | $2.51 \times 10^7$a ($9.33 \times 10^6$) | $1.13 \times 10^7$a ($1.70 \times 10^6$) | $1.00 \times 10^7$a ($2.18 \times 10^6$) |
| LAP | $1.26 \times 10^6$a ($2.30 \times 10^5$) | $9.86 \times 10^5$a ($1.77 \times 10^5$) | $8.48 \times 10^5$a ($1.30 \times 10^5$) |
| Enz. H′ | 0.007a (0.009) | 0.020ab (0.006) | 0.039b (0.007) |
| ETR | 1.06a (NA) | 19.21a (9.14) | 27.86a (11.06) |
| GPP | 0.27a (NA) | 4.80a (2.3) | 6.97a (2.8) |

Notes:
Standard error in parentheses except when missing data reduced $n < 3$ (NA). Lowercase letters indicate significant difference by ANOVA ($p < 0.05$).
EC, Electrical conductivity (μS/cm); Chl*a*, chlorophyll *a* (μg/g dry soil); Moist, gravimetric moisture (%); SOC, soil organic carbon (mg/kg dry soil); TN, total nitrogen (mg/kg dry soil); MBC, microbial biomass carbon (mg/kg dry soil); Bact. H′, TRFLP bacterial diversity; AG, α-glucosidase activity (nmol/g MBC/h); BG, β-glucosidase activity (nmol/g MBC/h); NAG, *N*-acetyl-β-glucosaminidase activity (nmol/g MBC/h); POX, phenol oxidase activity (nmol/g MBC/h); LAP, leucine aminopeptidase activity (nmol/g MBC/h); Enz. H′, carbon-acquiring enzyme diversity; ETR, electron transport rate (μmol/m²/s); GPP, gross primary production (μmol O₂/m²/s).

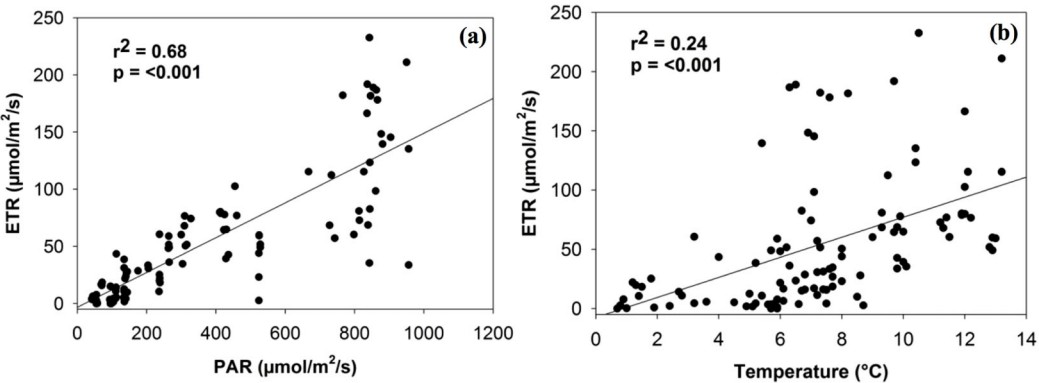

**Figure 2 Linear relationship between electron transport rate (ETR) and the density of photosynthetically active radiation (PAR) (A) and temperature (B).**

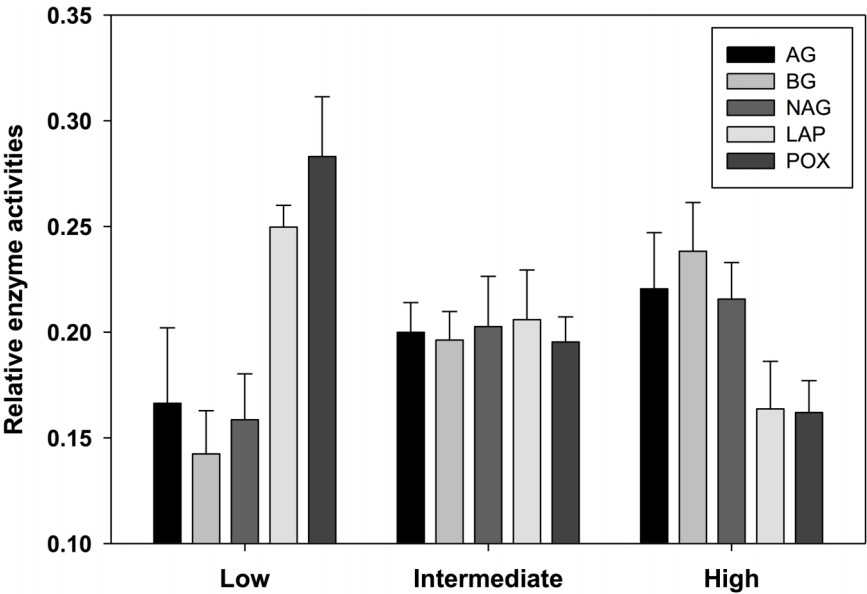

**Figure 3 Relative activity of five standardized ($(x - \text{mean})/\text{standard deviation} + 3$) exoenzymes for sample locations clustered into three productivity classes by soil moisture content.** The number of observations are $n = 3$ (low), $n = 7$ (intermediate), and $n = 5$ (high). AG, $\alpha$-glucosidase activity (nmol/g MBC/h); BG, $\beta$-glucosidase activity (nmol/g MBC/h); NAG, $N$-acetyl-$\beta$-glucosaminidase activity (nmol/g MBC/h); LAP, leucine aminopeptidase activity (nmol/g MBC/h); POX, phenol oxidase activity (nmol/g MBC/h).

productivity zones (e.g., low, intermediate, and high) to examine variability at this scale. Strong differences exist among productivity zones for many soil conditions such as pH and chlorophyll concentrations, and average values are reported per zone (Table 4).

PAM estimates of ETR were not significantly associated with soil moisture or chlorophyll *a*, although a positive trend did exist. ETR was significantly related to both PAR and temperature levels in a positive linear manner ($r^2 = 0.68$, $p < 0.001$; $r^2 = 0.24$, $p < 0.001$; Fig. 2). Although the relationship with PAR is largely expected (given that light intensity is a factor in calculating ETR), both relationships are consistent in magnitude with the findings of *Green et al. (1998, 2002)* for photobionts.

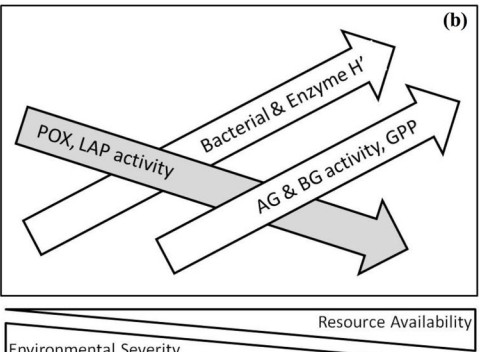

| | SLR Parameters ($r^2$) | | | | (a) |
|---|---|---|---|---|---|
| Factors | % Moist | Chl*a* | pH | EC | MLR ($R^2$) |
| AG | 0.36* | 0.16 | 0.14 | 0.27* | % Moist (0.36)* |
| BG | 0.50** | 0.50** | 0.21 | 0.43** | % Moist (0.50)** |
| POX | 0.35* | 0.31* | 0.26 | 0.33* | % Moist (0.35)* |
| LAP | 0.14 | 0.21 | 0.24 | 0.09 | pH (0.24) |
| Enz. H' | 0.34* | 0.29* | 0.32* | 0.29* | % Moist (0.34)* |
| Bact. H' | 0.34* | 0.25 | 0.01 | 0.09 | % Moist, pH (0.67)** |
| GPP | 0.39* | 0.15 | 0.05 | 0.38* | % Moist (0.39)* |

*$p < 0.05$; **$p < 0.01$; ***$p < 0.001$

**Figure 4** Simple linear regression (SLR, $r^2$) results (A) of soil factors against parameters associated with resource availability (e.g., % gravimetric moisture, chlorophyll *a*) and environmental severity (e.g., pH, electrical conductivity). Shaded cells indicate a negative relationship for simple regressions. Multiple linear regression (MLR, $R^2$) results indicate parameter(s) that best predicts soil factors. Illustration of SLR results (B) along a hypothetical environmental gradient. Moist, Gravimetric moisture (%); EC, electrical conductivity ($\mu$S/cm); Chl*a*, chlorophyll *a* ($\mu$g/g dry soil); AG, $\alpha$-glucosidase activity (nmol/g MBC/h); BG, $\beta$-glucosidase activity (nmol/g MBC/h); POX, phenol oxidase activity (nmol/g MBC/h); LAP, leucine aminopeptidase activity (nmol/g MBC/h); Enz. H′, index of activity for all carbon-acquiring enzymes; Bact. H′, TRFLP bacterial diversity; GPP, gross primary production ($\mu$mol O$_2$/m$^2$/s).

Soil enzyme activity varied significantly by location. POX and LAP (lignin- and protein-degrading enzymes, respectively) both exhibited a negative correlation with soil water content, as activity tended to be highest in the most arid habitats (Table 4). Activity of AG and BG (starch- and cellulose-degrading enzymes, respectively) had an opposite trend with the highest values found in productive soils, while NAG (chitin-degrading) activity exhibited no trend. An index of overall enzyme diversity was calculated using the Shannon–Wiener equation to highlight the relative change in evenness of carbon-acquiring enzyme activity (excluding protein-specific LAP), as described by *Tscherko et al. (2003)*. Enzyme diversity had a significant positive relationship with soil moisture ($r^2 = 0.34$; $p < 0.05$) and significant negative relationship with pH ($r^2 = 0.32$; $p < 0.05$). This result is a consequence of high POX activity (as indicated by the relative enzyme activity; Fig. 3) in arid soils, which is gradually replaced by a more even activity of all carbon-acquiring enzymes in productive soils. LAP activity did not correlate with total nitrogen concentrations, or either nitrate-N or ammonium-N (Data S1). Multiple regression results suggest soil water content as the driving force behind variation in most factors of biological diversity and activity (Fig. 4).

## DISCUSSION

Environmental gradients are a key feature of arid environments often chosen for investigation as the inferred mechanism underlying spatial patterns between productivity, for example, and diversity (*Noy-Meir, 1973*). While the diversity of microbial and metazoan communities in Antarctic terrestrial hotspots has been well characterized (*Simmons et al., 2009*; *Zeglin et al., 2011*; *Niederberger et al., 2015*), process-based measurements like primary productivity have received less attention. Here we contribute to this understanding of environmental gradients by quantifying the rates of certain

key functions that promote and reinforce the habitability of otherwise hyperarid Antarctic soils.

PAM fluorometry was used to measure the rate of PSII ETR, from which was calculated gross primary production (GPP). GPP, organic carbon, and chlorophyll *a* all peaked in the wettest soils that support the densest cryptogam mats. Average GPP was 6.97 $\mu$mol $O_2$/m$^2$/s in these most productive soils, with a maximum of 17.74 $\mu$mol $O_2$/m$^2$/s for one location. Assuming a 50% respiratory carbon loss during fixation (i.e., net primary productivity, NPP; *Schlesinger, 1997*) and 60 days of productivity per year (*Burkins, Virginia & Wall, 2001*), this is equivalent to an average NPP rate of ~217 g C/m$^2$/y with a maximum of 552 g C/m$^2$/y. Considering that some soils in Taylor Valley contain concentrations of organic matter approaching 250 g C/m$^2$ (*Moorhead et al., 2003*), mean residence time (pool/flux) of carbon in highly productive zones would equal ~2.2 years. This is substantially lower than the average residence time of decades to centuries estimated for soil organic matter in the broader Taylor Valley landscape (*Burkins, Virginia & Wall, 2001*; *Barrett et al., 2005*). Organic matter of productive dry valley soils is thus primarily labile photosynthates of recent origin that are rapidly utilized by soil decomposers; this situation is reversed with soil aridity, however, as soil carbon becomes increasingly dominated by recalcitrant substrates of ancient provenance (*Burkins, Virginia & Wall, 2001*).

Average GPP for the least productive soils was 0.27 $\mu$mol $O_2$/m$^2$/s. Again assuming that half of gross carbon and oxygen generation from GPP is consumed by respiration, this results in a NPP rate of 0.135 $\mu$mol $O_2$/m$^2$/s (or 0.135 $\mu$mol $CO_2$/m$^2$/s of autotrophic respiration). This falls within the range of total soil respiration rates previously described for arid dry valley soils (0.1–0.4 $\mu$mol $CO_2$/m$^2$/s; *Burkins, Virginia & Wall, 2001*; *Parsons et al., 2004*; *Ball et al., 2009*) and suggests that autotrophic respiration could constitute a substantial portion of total soil respiration even in arid soils. The consistency of these findings should encourage researchers to couple PAM fluorometry with soil $CO_2$ flux measurements in future work to attempt distinguishing rates of GPP, NPP, and respiration for producer communities at higher spatial and temporal resolution (*Pannewitz et al., 2006*). Recent evidence has also suggested that $CO_2$ efflux measurements from dry mineral soils in this region may be at least partially abiotic in origin (*Shanhun et al., 2012*; *Ball & Virginia, 2015*), which underscores the limit of $CO_2$ flux measurements to adequately depict (on its own) biological activity for this region. PAM fluorometry yields independent measures of primary production that can be used to further refine important properties of soil organic matter pools such as residence times.

Interestingly, while GPP inferred from PAM fluorometry appears to provide a valuable measure of soil productivity, it did not significantly correlate with chlorophyll levels (Table 3). This suggests that perhaps the better measure of soil productivity may ultimately depend on the temporal scale of inference. Concentrations of soil chlorophyll are thought to represent an integration of producer biomass accrual across days to weeks, from which can be inferred regular moisture availability and stable periods of producer growth. ETR and GPP values, however, provide rapid measures of

producer performance under instantaneous light, temperature, and moisture conditions that may not necessarily correlate with chlorophyll concentrations. The sensitivity of producers to rapid environmental change, identification of potential production-limiting stressors, and repeated (non-invasive) long-term measures of diel or even seasonal fluctuations in productivity of sample plots may be the best uses of PAM fluorometry. Indeed, previous work has applied PAM measures to reveal the effects of stress and damage to photosystem functioning (*Schreiber, 2004*).

Patterns in enzyme activity across the productivity gradient indicate distinct shifts in the nature of detrital pathways as well as organic matter pool complexity. Arid locations exhibited low evenness in an index of carbon-acquiring enzyme diversity, as indicated by the overall dominance of phenol oxidase activity (Fig. 3). This evidence suggests the soil organic matter pool in arid zones may be primarily composed of recalcitrant materials targeted by oxidative enzymes. Although vascular (lignin-bearing) primary producers are absent from this system, recalcitrant compounds may originate from the fatty acids and proteins of decomposing moss and lichen tissue (*Beyer et al., 1995*) deposited during ancient high-stands of a proglacial lake during the Last Glacial Maximum in Taylor Valley (*Hall, Denton & Hendy, 2000*). Such organic matter may be an important energetic source in the more hyperarid soils of this region (*Barrett et al., 2006a*). In a broader sense, the activity of oxidative enzymes worldwide has been found to be greater in drier, more alkaline soils, perhaps a consequence of high mineral surface stability (*Zeglin et al., 2009*), or higher phenol solubility under such conditions (*Sinsabaugh, 2010*). LAP was also relatively more active in arid rather than mesic locations, potentially indicating greater nitrogen limitation, reduced substrate (protein) availability, or potentially a higher enzyme reactive efficiency at increased pH as suggested by *Sinsabaugh et al. (2008)*. Recalcitrant organic substrates that we suspect to predominate in these arid soils, together with low nitrogen availability, likely contribute to the relatively low diversity and enzyme activity of microbial communities found in such habitats.

The relative activity of all carbon-acquiring enzymes (i.e., Enz. H′) indicated a more even carbon-acquiring enzyme diversity within productive soils (Table 4). From this we infer higher diversity of organic compounds in more productive regions, a logical conclusion considering the presence of greater producer biomass and diversity (*Orwin, Wardle & Greenfield, 2006*). Increased diversity of organic compounds may therefore be an additional factor behind the greater diversity of organotrophic bacterial communities in productive locations (*Grayston et al., 1998*). Niche differentiation among microorganisms for various substrates, particularly those that may be decomposed only via specialized enzymatic pathways, may be a mechanism (along with increased resource availability) responsible for increases in microbial diversity. Distinguishing the relative effects of resource quantity and quality remains an important direction for further research to establish important drivers of microbial community diversity.

The estimates of primary production we report, even within drier habitats, provide evidence that in situ carbon fixation is occurring widely across the McMurdo Dry Valley landscape with likely effects on subsurface communities and biogeochemical

rates. Environmental severity (soil pH, electrical conductivity) and resource availability (soil moisture, organic carbon concentration) vary inversely along a gradient of soil productivity and play important roles in determining biological diversity and activity, although moisture is the primary driver to explain community structure and function (Fig. 4). Changing enzyme activity along this gradient also highlights higher potential organic matter complexity in productive soils, an unforeseen factor that may promote microbial diversity. Our estimates for annual NPP of the most productive dry valley soils, colonized by mixed cyanobacteria and moss mats, indicate yields of ~217 g C/m$^2$/y, only slightly less than estimates of 250 g C/m$^2$/y for nearby coastal moss turfs (*Pannewitz et al., 2005*). We estimate annual NPP for the most arid soils (which dominate the dry valley landscape) to be ~8 g C/m$^2$/y, and thus an overall NPP for Taylor Valley soil productivity would be much lower than the global desert mean of 80 g C/m$^2$/y (*Waide et al., 1999*). In spite of these low rates a diverse and active organotrophic community persists here, a testament to the strength of interaction between ecosystem functioning (production, decomposition), environmental conditions (resource quantity/quality), and biotic diversity.

## ACKNOWLEDGEMENTS

We would like to thank the Crary Laboratory staff at McMurdo Station for their assistance, as well as Raytheon Company, Inc. and Petroleum Helicopters, Inc. for logistical support. We also thank Bobbie Niederlehner and several Virginia Tech collaborators for their contributions toward data acquisition and analysis.

### Funding

This research was funded by McMurdo LTER NSF OPP grant 1115245. There was no additional external funding received for this study. The funders had no role in study design, data collection and analysis, decision to publish, or preparation of the manuscript.

### Grant Disclosures

The following grant information was disclosed by the authors:
McMurdo LTER NSF OPP: 1115245.

### Competing Interests

The authors declare that they have no competing interests.

### Author Contributions

- Kevin M. Geyer conceived and designed the experiments, performed the experiments, analyzed the data, contributed reagents/materials/analysis tools, wrote the paper, prepared figures and/or tables, and reviewed drafts of the paper.
- Cristina D. Takacs-Vesbach conceived and designed the experiments and reviewed drafts of the paper.

- Michael N. Gooseff conceived and designed the experiments and reviewed drafts of the paper.
- John E. Barrett conceived and designed the experiments, contributed reagents/materials/analysis tools, and reviewed drafts of the paper.

## Field Study Permissions

The following information was supplied relating to field study approvals (i.e., approving body and any reference numbers):

Field sampling was permitted under McMurdo LTER NSF OPP grant 1115245.

## Data Availability

The raw data has been supplied as Supplemental Dataset Files.

## Supplemental Information

Supplemental information for this article can be found online at http://dx.doi.org/10.7717/peerj.3377#supplemental-information.

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
