# Peer review of "Primary productivity as a control over soil microbial diversity along environmental gradients in a polar desert ecosystem"

_PeerJ, doi:10.7717/peerj.3377_

## Round 0.1 · original submission · Major Revisions

Dear Kevin, first of all I apologize for the delay, we had trouble getting reviewers familiar with PAM. All 3 reviewers and I thought that the manuscript was an important contribution to the field and highlights a potentially innovative approach to linking terrestrial productivity to patterns of microbial diversity. As you will see the only really substantive issue is with benchmarking of the PAM results (particularly the influence of water on instantaneous PAM measurements), plus a response is needed about enzyme assay controls. Most comments were minor but I have selected 'Major Revisions' as it was not clear to me whether your response requires additional work to determine the validity of the PAM measurements or whether you have the data in hand already to support that. I hope this is straightforward to resolve.

·

Basic reporting

No comment

Experimental design

No comment

Validity of the findings

No comment

Additional comments

The authors present a series of measurements estimating photosynthetic (and therefore C-fixing) activity in the McMurdo Dry Valleys along a moisture gradient from streams. Such measurements are very important for understanding the C cycle and basal food resources for this ecosystem where it has traditionally been very difficult to measure productivity. They introduce the use of PAM for tackling this difficulty. The manuscript is straightforward, clearly-written, and succinct. I provide only a minor comment:

Lines 211-213/Table 4: Can you be explicit in how you defined these three zones? Obviously it’s by soil moisture, which are listed for each zone in the table, but how did you decide on the cutoff points?

·

Basic reporting

The manuscript is written clearly and concisely, and the results were well contextualized. The title is very straightforward and accurately represents the work. The only comment I have regarding the presentation of the manuscript is that there should be sub-headings in Site Description and Methods to improve readability.

Experimental design

The research question is interesting and fills a well-articulated gap in knowledge for the Dry Valley ecosystem. Broadly speaking, the findings are logical and congruent with expectations based on existing knowledge. However, there are two issues with experimental design that lessen the robustness of the findings:
1. The lack of reliable points of comparison for the PAM measurements
2. The lack of kill/inhibitor controls for enzyme activity measurements
As far as this reviewer is aware, there have been few reliable reported measurements of primary productivity using PAM on organics-poor soils. Although PAM measures a very specific phenomenon (drop in photochemical quenching of fluorescent signals), it cannot be assumed that readings obtained by PAM are entirely associated with photosynthesis when there is very low chlorophyll content in the substrate being characterized. Note that I'm not say that the readings are false, but it is difficult to quantify exact how much estimated photosynthetic activity is actually biological. It would have been highly useful to have control measurements of bedrocks and active cyanobacterial mats or mosses to contextualize the readings reported here. Another issue with PAM is that it is an instantaneous measurement, which means the readings are much more reflective of the limiting factor for photosynthesis (i.e., water) than capacity for photosynthesis. This is why water augmentation is a standard procedure when PAM is used to characterize lichen physiology since the capacity for photosynthesis is what determines the primary production capability of a system, not moisture content at the time of sampling. This potentially explains why ETR and GPP values did not correlated with ChlA levels (lines 264-265).
A similar issue exists for the enzyme assay data since they (i.e., AG and POX) appear to correlate with moisture content. Again, the data make sense and are likely to be (at least partially) derived from enzymatic activities, but there is not enough confidence without kill/inhibitor controls to conclude that all of the activities observed are in fact due to enzymes released by resident microorganisms.

Validity of the findings

As stated above, the findings are logical and in line with what can be reasonably expected from existing knowledge. However, I do not think the experimental design allows the PAM and enzyme activity data to be unequivocally attributed to resident microorganisms. Without knowing how much of the PAM readings is attributable to primary production, I would feel much more comfortable using the GPP and NPP as upper limits rather than estimates. The application of Shannon diversity index on enzyme activity data is unusual and likely violates the statistical assumption of the index (i.e., each "species" or "character" being independent of each other) since it is likely some microorganisms possess multiple of the enzymes assayed. I would also suggest that the authors be much more explicit and careful when discussing parameters that likely auto-correlate (e.g., MBC vs. TN and SOC).
A few specific comments:
• Lines 215-217: Is ETR related to PAR or not? These two sentences appear to contradict each other.
• Lines 243-245: Please provide citations or rationale for the 50% respiratory carbon loss and 60 growth (not growing) day estimates.
• Lines 253-255: I'm puzzled by the comparison between NPP figures and soil respiration rates since they are associated with two distinct microbial populations (photoautotrophs vs. heterotrophs). I should also note that recent evidence (in addition to the two papers cited later in the paragraph) indicates that most, if not all, of the observed CO2 flux in the Dry Valleys are abiotic in nature. Much more work is required before primary and secondary productivity can be measured directly for the Dry Valleys.
Lines 275-277: Please elaborate on what low evenness of Enz. H' means ecologically.

Additional comments

Overall, I think the research question is important and interesting, but the results are perhaps not as quantitative as the authors have implied or assumed. Given the logistical difficulties associated with this research, it is unreasonable to expect any revision or repetition of the experimental work. However, this reviewer suggests that the complications associated with experimental design be acknowledged in results and considered when interpreting the data in discussion. A revised manuscript containing those changes would be suitable for publication.

·

Basic reporting

The manuscript was clearly articulated and described the use of PAM fluorometry to examine primary productivity rates in Dry Valley, Antarctica soils of Taylor Valley.

General comments:
The use of PAM fluorometry in conjunction with enzyme assays is a useful addition to the literature on microbial activity in hyper-arid Antarctic permafrost soils, specifically primary productivity for which there is limited data.


Specific comments:
Photos of cryptograms sampled would be useful for this manuscript.

ln 94. The assumption that biological activity in Taylor Valley is limited to a 8 week period in austral summer is likely incorrect. Bakermans et al. 2014 (doi: 10.1111/1574-6941.12310) demonstrated that heterotrophy occurs in soil samples from Taylor Valley at -5 C. Additionally, the high salt content described in these soils (ln 93) aid in the formation of liquid water at colder temperatures, the likely limiting factor for activity.

Need more details on the sites collected along the environmental gradient. For example.

How do these regions vary ( pH, moisture content etc). In a broader context - how do they compare to other Dry Valley sites for which microbial activity has been detected (ex- Taylor Valley) or not detected (ex- University Valley)

ln. 124. In table 1 or in text, please specifically describe which samples correlate to the described env. conditions ( ex- which site were the hyperarid soils collected from ?).

ln 126. Describe the locations in each region.


ln 174. Missing word Amplitude.

ln 215. Change to: "..ETR were not significantly associated with soil moisture and chlorophyll a.."

ln 217. P-value?

ln 243. Please justify why the assumption of 50% respiratory carbon loss.



ln 244. Again, note that communities may very well be growing for longer than 60 days annually.

Ln 264. Since GPP is calculated as 1/4th of ETR, please remove one of these terms from the sentence (ie- ETR is correlated, or GPP is correlated; not GPP and ETR is correlated).

ln 288-291. Describe how the TRFLP data reflects this.

Experimental design

The choice for sites should be better described in the manuscript (more detail needed).


How were PAM measurements taken (more detail needed). Were cryptograms targeted specifically? Where samples dark covered first?

Validity of the findings

no comment.

---

## Round 0.2 · accepted · Accept

Dear Kevin, I shared your response with the reviewer and they were happy to accept the manuscript in this form. Congrats on a really nice manuscript and thank you for your patience during this process.

Regards - Eoin

·

Basic reporting

The manuscript continues to be well-written and succinct. Clarity has been further improved due to revisions based on reviewer comments, particularly regarding the PAM measurements.

Experimental design

The experimental design seems appropriate for the interpretation of the results.

Validity of the findings

The results are supported by the data available. Dr. Lee brought up a good point about the PAM measurements essentially integrating photosynthetic capacity vs. resource limitation in this case, and in addressing this the authors improve the results & discussion. The way I read it, the authors are interpreting the PAM results as an ecosystem processes, essentially regardless of whether it's capacity or water resources that limit it, rather than claiming to measure the capacity for GPP. (And I think the added text in the discussion is an interesting little tidbit about short vs long-term processes.) So given the scope of the interpretation, I think the results are supported by the experimental design.

Additional comments

I have no further comments.

·

Basic reporting

No comment

Experimental design

Just because kill controls have not been included in the standard protocol does not mean they are not necessary. The concern is not with magical abiotic breakdown of organic materials, but with the stability of the substrates used in the assays. For example, L-DOPA used for the phenol oxidase assay is known to be unstable at high pH (which Dry Valley soils typically are). Without data from kill/inhibitor controls, it is imprudent to interpret enzyme activity data quantitatively.

Validity of the findings

Following my reasoning above, it is not clear to me that calculating diversity indices for enzyme activities and assigning ecological relevance to them is a good idea since the enzyme activity data cannot be assumed to be quantitative.

Additional comments

No comment

·

Basic reporting

No comment

Experimental design

No comment

Validity of the findings

No comment

Additional comments

The authors have made adequate minor revisions for publication